# Idiopathic Penile Calcinosis Cutis: A Histopathological Case Report

**DOI:** 10.3390/reports8040248

**Published:** 2025-11-27

**Authors:** George Stoyanov, Dobri Marchev, Hristo Popov

**Affiliations:** 1Department of Pathology, Multiprofile Hospital for Active Treatment, 9700 Shumen, Bulgaria; 2Department of Urology, Multiprofile Hospital for Active Treatment, 9700 Shumen, Bulgaria; 3Department of General and Clinical Pathology, Forensic Medicine and Deontology, Faculty of Medicine, Medical University—Varna, 9002 Varna, Bulgaria

**Keywords:** calcinosis cutis, idiopathic calcinosis cutis, genital calcinosis cutis, penile calcinosis cutis

## Abstract

**Background **** and Clinical Significance**: Calcinosis cutis is a rare condition that can develop through several mechanisms. These include dystrophic, calciphylaxis (classical, metastatic, and iatrogenic), and idiopathic mechanisms. Idiopathic calcinosis cutis is rare and always a diagnosis of exclusion. A particularly rare site for the development of idiopathic calcinosis cutis is the penis. **Case Presentation**: A previously healthy 18-year-old male presented to our institution with a three-month history of a painless, firm swelling on the outer layer of the prepucium in the area of the commissure. Histopathology of the excised specimen showed a varying caliber of calcium deposits within the dermis, ranging from small psammoma-like bodies to larger calcium deposits measuring up to 2.5 mm. The deposits were freely dispersed within the dermal collagen and did not exhibit vascular affinity, nor surrounding foci of inflammation. The epidermis was not involved, with only mild reactive hyperkeratosis. The results of detailed physical, imaging, and laboratory tests were normal, and hence the diagnosis of idiopathic calcinosis cutis of the penis was established. **Conclusions**: Penile calcinosis cutis is a rare condition that falls within the broader group of genital calcinosis cutis. The condition is typically present in young males and has an excellent prognosis after excision.

## 1. Introduction and Clinical Significance

Calcinosis cutis, by definition, is a condition involving the deposition of insoluble calcium salts (deposit) within the dermis [1]. There are several main types of this condition, defined by their underlying etiopathogenic mechanisms. The first is dystrophic, in which the cause of calcium deposition is chronic tissue injury with incomplete regeneration, mainly observed in trauma, inflammatory conditions, and collagenoses (dermatomyositis, lupus, and systemic sclerosis) [1,2]. Often, dystrophic calcinosis cutis is also be associated with cutaneous tumors such as calcifying pilomatrixoma (Malherbe tumor), trichoblastomas, and basal cell carcinoma [1,3].

The second type is the largest group of calciphylaxis conditions, including classical calciphylaxis and iatrogenic and metastatic calcinosis. For these to develop, serum calcium and phosphate levels need to be elevated, leading to tissue precipitation of insoluble calcium salts [1]. For classical calciphylaxis to occur, the typical causative agent is the presence of severe and chronic kidney disease, characterized by the significant dysregulation of calcium and phosphate metabolism and homeostasis, leading to the predominantly vascular deposition of calcium and secondary ischemic changes within the tissue, with further deposition of calcium, with the possibility of subcutaneous involvement as well [4]. Metastatic calcification is closely linked in its mechanism to calciphylaxis, and the two conditions can often coexist, presenting a spectrum of conditions. However, the deposits are predominantly extravascular and can accumulate in extracutaneous sites, primarily in the muscles, joint soft tissues, fascia, and rarely in the internal organs [1,4]. Common causative agents of metastatic calcinosis, in addition to chronic kidney injury, include hypervitaminosis D and hyperparathyroidism, underscoring the role of the endocrine system in this spectrum of conditions [1,4]. Iatrogenic calcinosis develops via the same pathophysiologic mechanisms as calciphylaxis and metastatic calcinosis; however, the underlying reason is not the dysregulation of metabolism and homeostasis, but rather the administration, typically intravenously and at a high concentration in a short administration time frame, of calcium- and phosphate-containing drugs [1,5,6]. Keloidal cutaneous scars with calcification, despite some of them being considered iatrogenic, should not be viewed in this category, but rather in the dystrophic one [1,7].

The last category of calcinosis cutis is the idiopathic category, for which the morphological findings are the same, with calcium deposits within the dermis; however, in this category no underlying cause can be identified [8,9]. Hence, idiopathic calcinosis cutis is a diagnosis of exclusion.

There are several exceedingly rare sites for calcinosis cutis; one of these is the genital area, known as genital calcinosis cutis, which predominantly affects the scrotum in males and the labia majora in females [10]. Herein, we present a case of one of the rare subtypes of idiopathic calcinosis cutis—penile calcinosis cutis.

## 2. Case Presentation

A previously healthy, uncircumcised 18-year-old male presented to our institution with a three-month history of a painless, firm swelling on the outer layer of the preputium in the area of the commissure. The patient denied experiencing previous trauma to the region or engaging in risk behaviors. The physical exam revealed a 5 mm, relatively freely mobile, painless lesion in the depicted area. Given the suspicion of a granuloma, a local excision was recommended, and the patient agreed. The procedure was performed under local anesthesia and proceeded without complications, with an uneventful post-intervention period. The excised specimen was sent for histopathology.

The specimen itself comprised leaf-like skin measuring 14 × 9 mm post-fixation, with a slightly elevated central area. This area, when sectioned, measured 8 mm at its largest and had a chalky appearance in the cross-section.

Histopathology showed that the resection margins were free from pathological processes. The grossly chalky lesion was represented by dermal lesions with varying-caliber calcium deposits, ranging from small psammoma-like bodies to larger deposits measuring up to 2.5 mm (Figure 1). The deposits were freely dispersed within the dermal collagen and did not exhibit vascular affinity, nor surrounding foci of inflammation. The epidermis was not involved, with only mild reactive hyperkeratosis (Figure 1).

Penile calcinosis cutis was established and a detailed clinical history and laboratory tests were performed to exclude non-idiopathic causes. The patient once again denied engaging in risk behaviors or experiencing previous trauma, and did not have a significant history of medication use; imaging modalities showed a normal anatomy, and the bloodwork and urinalysis were completely within the reference range. Hence, the diagnosis of idiopathic penile calcinosis cutis was accepted.

One year after the intervention, the patient is regular in following up, and no new sites or suspected areas of cutaneous calcinosis have been noted.

## 3. Discussion

Idiopathic calcinosis cutis is a rare condition, primarily a diagnosis of exclusion [1]. An extremely rare site for calcinosis cutis is the genital area, designated as genital calcinosis cutis, which primarily affects the labia majora in females and the scrotum in males (scrotal calcinosis cutis) [10,11].

Genital calcinosis cutis in males predominantly develops in younger patients. However, cases of older males have also been described, typically with a more advanced state at the time of presentation, underlining the potential for a chronic and progressive nature of the condition [11,12]. As with most other sites of calcinosis cutis, surgical excision or extirpation is typically curative, and the prognosis in idiopathic cases is excellent, with low risk of recurrence [10,11,12].

The etiopathogenesis of genital and scrotal calcinosis cutis is widely debated, and in recent times, a broader discussion has emerged regarding the dystrophic nature of the condition, namely the degeneration and secondary calcification of predominantly epidermal and other keratocysts [10,12,13]. This mechanism seems highly plausible, as epidermal cysts in the genital area are common; however, the complete calcification of keratocysts, without evidence of at least partial retention of the epithelial lining or granulomatous inflammation, is rare [14,15,16]. Alternatively, the idiopathic nature of genital calcinosis may be similar to that of osteoma cutis and could represent an atavism, especially in extensive cases [11,17].

However, the dystrophic nature of the calcification of keratocysts seems highly unlikely in calcinosis cutis of the penis, as both the shaft and prepucium lack skin appendages, which are crucial for the development of some of these lesions [14,18].

Penile cases of calcinosis cutis are an extremely rare occurrence, of which only a handful of cases have been described [10,19,20,21,22,23]. The condition predominantly affects younger males, as seen in the currently depicted case [10,19,20,21,22,23]. As with other locations of calcinosis cutis, a detailed differential diagnosis is required to exclude other, more common causes, such as calciphilaxis, which is prevalent in penile locations [23]. However, penile calciphylaxis is always present in the background of a severe underlying condition and typically develops in older males [23]. Clinical presentation between the two conditions further varies: calciphylaxis is typically painful and often concomitant with skin ulceration, whereas calcinosis cutis is painless and has an intact epidermis [10,13,19,20,21,22,23]. Furthermore, unlike in penile calciphylaxis, where the prognosis is typically poor due to the underlying condition, penile calcinosis cutis usually has an excellent prognosis with low recurrence rates after surgical intervention [10,13,19,20,21,22,23].

Alternatively, the dystrophic nature of the condition can again be argued as being based on inflammation and chronic irritation, especially in individuals engaging in risk behavior. However, in the reported cases in the literature, most patients are young and do not report engaging in risk behavior, eliminating both potential mechanisms and the timeframe required for calcinosis cutis to develop [10,13,19,21,22]. One reported case in the literature presented with both scrotal and penile calcinosis cutis, underscoring the possibility of a shared driving mechanism between the two conditions [19].

To the best of the authors’ knowledge, the presented case report is only the sixth in the literature to depict idiopathic calcinosis cutis in the penile area [19,20,21,22].

## 4. Conclusions

Idiopathic penile calcinosis cutis is a rare condition in the spectrum of genital calcinosis cutis. The condition, while having a favorable outcome, should come as a diagnosis of exclusion based on the myriad of progressive conditions that can have a similar manifestation and mechanism when compared to it, such as penile calcyphilaxis.

## Figures and Tables

**Figure 1 reports-08-00248-f001:**
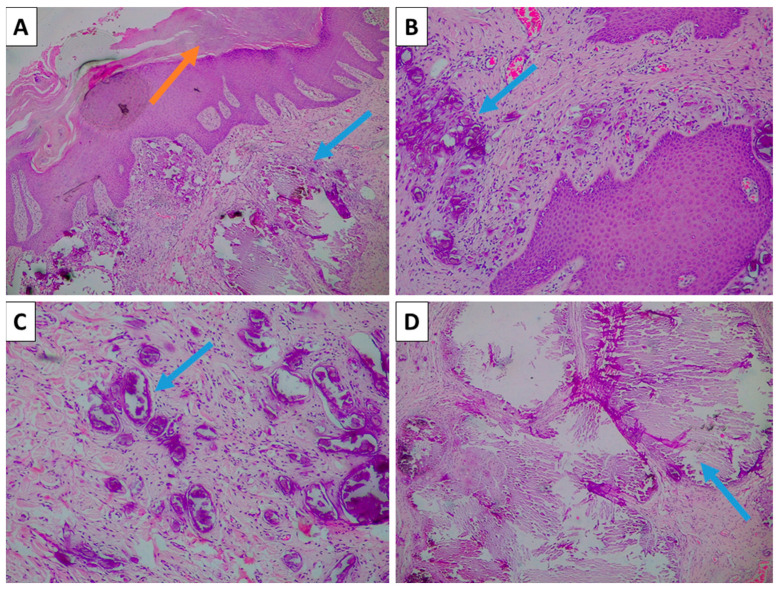
Histopathology of the lesion. (**A**) Mildly hyperkeratotic epidermis (orange arrow) and dermal calcium deposits (blue arrow), H&E stain, original magnification 40×; (**B**) small, densely aggregated psammoma-body-like calcium deposits (blue arrow), H&E stain, original magnification 100×; (**C**) small but varying-caliber psammoma-body-like calcification without vascular wall affinity or surrounding inflammation (blue arrow), H&E stain, original magnification 100×; (**D**) larger-caliber calcification (blue arrow), H&E stain, original magnification 40×.

## Data Availability

The original contributions presented in this study are included in the article. Further inquiries can be directed to the corresponding author.

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
