# Peer review of "Idiopathic Penile Calcinosis Cutis: A Histopathological Case Report"

_reports, 2025, doi:10.3390/reports8040248_

Round 1
Reviewer 1 Report
Comments and Suggestions for Authors Dear Author; Calcinosis cutis is a well-known skin condition. While scrotal calcinosis cutis is more common, penile calcinosis cutis is also reported in the literature. This topic, with published case series, is not uncommon enough to contribute to the literature. We can consider it a well-written article, but I don't believe it will make a significant contribution to the literature. sincerelyAuthor Response
Reviewer 1:
Dear Author; Calcinosis cutis is a well-known skin condition. While scrotal calcinosis cutis is more common, penile calcinosis cutis is also reported in the literature. This topic, with published case series, is not uncommon enough to contribute to the literature. We can consider it a well-written article, but I don't believe it will make a significant contribution to the literature. Sincerely
- Dear reviewer, thank you for your comments. Calcinosis cutis is considered a rare dermatological condition; the form depicted in the presented case is idiopathic and occurs in a young adult. Given the location, this is an extremely rare occurrence (see literature references in the manuscript). While there have been reports of calcinosis cutis of the penis, most of these reports and case series focus on calciphylaxis, which is observed in the elderly and in the context of CKD (see literature references in the manuscript). To our knowledge, there have been no more than five case reports of idiopathic calcinosis cutis of the penis and no case series published on the topic. This has been reflected upon in the discussion section of the text.
- We have no additional compelling comments to persuade the reviewer of the rarity and significance of the text, other than it being the sixth (to the best of our knowledge) reported case in the literature.
Reviewer 2 Report
Comments and Suggestions for Authors
Some spelling mistakes:
Line 96: risk (not kisk).
Line 130: calciphylaxis ( not calciphilaxis).
Comma sign missing in some paragraphs.
It this form of idiopathic calcinosis cutis equivalent with subepidermal calcified nodules?
Comments on the Quality of English LanguageI think some comma signs are missing.
Author Response
Reviewer 2:
Comments and Suggestions for Authors
Some spelling mistakes:
Line 96: risk (not kisk).
Line 130: calciphylaxis ( not calciphilaxis).
Comma sign missing in some paragraphs.
- Dear reviewer, thank you for noticing these typographical errors on our behalf. Revisions will be made to the language and grammar.
It this form of idiopathic calcinosis cutis equivalent with subepidermal calcified nodules?
- Calcinosis cutis in the present form refers to the aggregation of calcium salts with no underlying cause. Calcified nodules can be viewed as a type of calcinosis cutis; however, in that instance, they are caused by an evident underlying condition such as infection, trauma, foreign body, or a vascular condition.
Comments on the Quality of English Language
I think some comma signs are missing.
- Revisions on language and grammar will be made.
Reviewer 3 Report
Comments and Suggestions for Authors
line 35 ("collagenases"): collagen vascular disorders (would be more appropriate)
line 96 ("patient... denied kisk [?] behaviour."). Please rephrase or explain.
Author Response
Reviewer 3:
ne 35 ("collagenases"): collagen vascular disorders (would be more appropriate)
- Typographical error has been corrected, vasculitis are only one type of collagenoses, so this suggestion has not been implemented
line 96 ("patient... denied kisk [?] behaviour."). Please rephrase or explain.
- Thank you for noticing this typographical error on our behalf. Risk behavior has been corrected
Reviewer 4 Report
Comments and Suggestions for Authors I believe this is a highly relevant article, as case reports are notonly useful for medical knowledge, but also humanize science by
highlighting conditions that might be overlooked due to their low frequency.
We could even consider including them among rare diseases.
Knowledge of these conditions could help reduce diagnostic errors,
and their correct identification and understanding could allow the use
of imaging alternatives such as infrared tomography to determine
calcifications. Similarly, other therapeutic alternatives,
such as chelating agents, could be used as an alternative to current
surgical options.
Author Response
Reviewer 4:
I believe this is a highly relevant article, as case reports are not
only useful for medical knowledge, but also humanize science by
highlighting conditions that might be overlooked due to their low frequency.
We could even consider including them among rare diseases.
Knowledge of these conditions could help reduce diagnostic errors,
and their correct identification and understanding could allow the use
of imaging alternatives such as infrared tomography to determine
calcifications. Similarly, other therapeutic alternatives,
such as chelating agents, could be used as an alternative to current
surgical options.
- Dear reviewer, thank you for these kind comments. However, we do not delve into fields we are not specialized in, such as the aforementioned imaging modalities. Furthermore, given the rarity of the condition — to the best of our knowledge, this is the sixth reported case in the literature — it would be valuable for additional cases to be published. Future detailed reviews or meta-analyses could explore the condition further and help draw more definitive conclusions on these topics.
Round 2
Reviewer 1 Report
Comments and Suggestions for Authors
Dear Authors,
It is well written article but It's not a rare or interesting topic. It's a lesion we routinely encounter among skin lesions. Scrotal lesions are common. However, there are also case series on penile lesions. I don't think it will contribute to the journal or the literature. Best regards.
Reviewer 3 Report
Comments and Suggestions for Authors
".....ne 35 ("collagenases"): collagen vascular disorders (would be more appropriate)
- Typographical error has been corrected, vasculitis are only one type of collagenoses, so this suggestion has not been implemented...."
I would be more precise with current terminology:
CONNECTIVE TISSUE DISEASES would be more appropriate for the international readership, as "Kollagenosen", "collagenoses" is a more or less central European term - which might lead to significant misunderstanding in anglo-saxon countries, e.g. WEEDON uses the term COLLAGENOSIS for completely different alterations (collagenosis nuchae, fibroblastic rheumatism and many others)